# Insights into the Peritumoural Brain Zone of Glioblastoma: *CDK4* and *EXT2* May Be Potential Drivers of Malignancy

**DOI:** 10.3390/ijms24032835

**Published:** 2023-02-02

**Authors:** Martina Giambra, Andrea Di Cristofori, Donatella Conconi, Matilde Marzorati, Serena Redaelli, Melissa Zambuto, Alessandra Rocca, Louis Roumy, Giorgio Carrabba, Marialuisa Lavitrano, Gaia Roversi, Carlo Giussani, Angela Bentivegna

**Affiliations:** 1School of Medicine and Surgery, University of Milano-Bicocca, 20900 Monza, Italy; 2PhD Program in Neuroscience, University of Milano-Bicocca, 20900 Monza, Italy; 3Division of Neurosurgery, Fondazione IRCCS San Gerardo dei Tintori, 20900 Monza, Italy

**Keywords:** glioblastoma, peritumoural brain zone, *CDK4*, *EXT2*, malignant transformation

## Abstract

Despite the efforts made in recent decades, glioblastoma is still the deadliest primary brain cancer without cure. The potential role in tumour maintenance and progression of the peritumoural brain zone (PBZ), the apparently normal area surrounding the tumour, has emerged. Little is known about this area due to a lack of common definition and due to difficult sampling related to the functional role of peritumoural healthy brain. The aim of this work was to better characterize the PBZ and to identify genes that may have role in its malignant transformation. Starting from our previous study on the comparison of the genomic profiles of matched tumour core and PBZ biopsies, we selected *CDK4* and *EXT2* as putative malignant drivers of PBZ. The gene expression analysis confirmed their over-expression in PBZ, similarly to what happens in low-grade glioma and glioblastoma, and *CDK4* high levels seem to negatively influence patient overall survival. The prognostic role of *CDK4* and *EXT2* was further confirmed by analysing the TCGA cohort and bioinformatics prediction on their gene networks and protein–protein interactions. These preliminary data constitute a good premise for future investigations on the possible role of *CDK4* and *EXT2* in the malignant transformation of PBZ.

## 1. Introduction

Glioblastoma (GBM) is the most common primary brain tumour with an aggressive behaviour and a poor prognosis that is due to cellular high proliferation rate and to high resistance to adjuvant treatments [1,2,3]. Nowadays, first line treatment consists of maximal safe surgical resection followed by adjuvant combined radiotherapy and chemotherapy with temozolomide. With this combination of treatments, patients have a median overall survival of 12–16 months that can be increased with the addition of tumour treating fields (TTF) according to recent literature [4]. Death is due to tumour recurrence that occurs in more than 90% of cases [4,5]. Clinical and orthotopic models have shown that the typical growing pattern of GBM is characterized by infiltration of the peritumoural brain zone (PBZ), the apparently healthy marginal brain tissue surrounding the tumour [6,7,8,9]. The PBZ is defined on brain magnetic resonance imaging (MRI) as the non-contrast-enhancing peritumoural region of the GBM and it contains several cellular populations such as tumour infiltrating cells, glioma-associated stromal cells, and vascular cells [3]. In recent years, efforts have been made to examine cellular, molecular, and genetic features of such microenvironments, and its interaction with the tumour core (TC), since it is known that tumour recurrence often occurs in the context of PBZ [4,5,10]. Nevertheless, sampling normal brain tissue is not always possible due to the eloquent location of many GBMs or due to the ethical issues related; therefore, deep and wide studies on the PBZ are still few [3]. Nowadays, it is known that PBZ contributes to shaping the tumour microenvironment, enhancing tumour self-renewal, and promoting the infiltration of the neoplastic cells [3,11,12,13]. Furthermore, molecular investigations have revealed a significant level of genomic instability, not only in the TC but also in the PBZ [3,10,11,12,13]. The oncogenic properties of the PBZ are further supported by the fact that supramarginal resection has a positive impact on overall survival (OS) of some patients, although without definitively avoiding tumour recurrence [14,15]. However, the biological role of the PBZ is not still completely understood and the molecular mechanisms that carry its oncogenic features are not elucidated. For this reason, in this study, starting from the comparison of the genomic profiles of matched TC and PBZ biopsies, possible PBZ genes of interest were selected, their expression profiles were correlated with patients’ outcomes and their possible role in GBM was assumed by consulting databases. 

## 2. Results

### 2.1. Gross Total Tumour Resection Affects Patients’ Overall Survival

Table 1 summarizes the demographic data and the clinic–pathological features of the enrolled patients. It is also reported in which cases the collection of the PBZ was possible and the disease monitoring and surveillance of patients. Firstly, the reported features were analysed in order to assess which factors influence the survival of the cohort. A total of 20 out of 28 patients (71.4%) received a gross total resection (GTR) of contrast-enhancing tumour. A total of 2 patients (patient 14 and 30) died due to post-operative complications and they were not considered for overall and disease-free survival analysis. A total of 12 out of 26 patients (46.15%) were alive at the end of the study (median overall survival of 11.5 months). The average age onset of alive patients was 59 years old (Table 1). 

A total of 14 (54%) out of 26 of patients died before the end of the study and their outcome correlated with an older age (mean 64.7 years old). Moreover, taking into account the extent of surgical resection (EOR), the median OS was 16 and 14 months for patients who received GTR and sub-total resection (STR), respectively. Patients with GTR had a significant higher OS when compared with patients with a subtotal one (hazard ratio: 2.854, confident interval: 0.6926 to 11.76, *p* = 0.047) *(*Figure 1a). The small sample size did not allow finding a statistical difference in terms of disease-free survival, although there was a trend for better DFS among patients who received a GTR (Figure 1b). In addition, the methylation status of the promoter of the methylguanine methyltransferase (MGMT) protein did not show an association with the follow-up outcomes of the cohort, without a benefit in OS or DFS (Table 1) (Fisher’s exact tests *p* > 0.05). 

These data reveal that GTR influenced the overall survival and not the disease-free survival, but this finding may be related with the small sample size. The OS is also influenced by patients’ age onset.

### 2.2. Genes Involved in Copy Number Gains Were Also Transcriptionally Over-Expressed in PBZs

In order to identify genes imbalanced in the PBZ and potentially associated to its malignant transformation, genomic analysis was performed on 15 matched TC–PBZ specimens (Table 1). A total of 10 pairs were included in our recent work [11], while 5 pairs were still unpublished. The Pearson correlation metrics were calculated on the new matched biopsies, as already performed in our previous work [11] (Table 2). Regarding the new TC–PBZ pairs, 1 from patient 38 had a moderately strong correlation between the two genomic profiles (R = 0.67). Chromosome 7 amplification, the canonical alteration of GBM, was found in the TC and also in the PBZ, confirming the infiltration of cancer cells in this area. A total of three out of five pairs had low correlation, highlighting the presence of exclusive CNAs of the PBZ. Meanwhile, 1 out of the 5 pairs (from patient 33) did not report any CNA at all. 

Starting from the comparison of the matched genomic profiles and applying the two strategies described in Section 4, six CNAs affecting the PBZ were identified and reported in Table 3. Among the selected CNAs, only copy number gains were arbitrarily picked out with the aim of identifying new oncogenes. Among the genes included in the acquired regions, *CDK4* and *EXT2* were finally selected after consulting the literature and databases (Table 4).

Then, the expression level of the selected genes was assessed in 10 PBZ biopsies from patients who received GTR, since genomic imbalances did not always correlate with the altered transcriptional level of the involved gene. *CDK4* and *EXT2* showed higher transcriptional levels in the PBZ than in the control (NHA cells) in six and nine cases, respectively (Figure 2).

Six biopsies showed a co-over-expression, while in the biopsies from patient 6 the two genes were lower expressed than in the control. In three PBZs, the over-expression of *EXT2* did not correlate with the *CDK4* one. Interestingly, increased expression levels of the selected genes were seen in both biopsies bearing CN gain and in ones where the genomic analysis did not show copy number imbalance, such as the biopsy from patients 9, 15, and 33 concerning *CDK4*, and patients 10, 15, 17, 20, 24, 33, and 38 concerning *EXT2*.

### 2.3. Potential Role of CDK4 and EXT2 in the Malignant Transformation of the PBZ

In order to evaluate the potential role of the selected genes in the malignant transformation of the PBZ, the expression levels of *CDK4* and *EXT2* in both the studied cohort and in the databases’ cohorts were correlated to survival analysis. Since in the current cohort, four patients had *CDK4* lower expressed in PBZ than in the normal control and in six patients it was vice versa, to determine whether the *CDK4* higher expression might be an index of malignant transformation of this area and therefore be a negative prognostic marker, survival analysis was performed comparing the two groups. An increased expression of *CDK4* in the peritumoural margin of GBM seemed to reduce the overall survival of these patients, as observed in Figure 3 (hazard ratio: 5.278, confident interval: 0.8991 to 30.99, *p* = 0.050). Regarding *EXT2*, most of the patients (90%) had higher expression in the PBZ than in the normal control (Figure 2), thus it was not possible to classify them into two groups and survival analysis was not performed. 

Next, to validate the assumed role of *CDK4* and to deepen *EXT2*’s one, we used a bioinformatics approach. The available databases did not include the peritumoural brain zone, so we focused our analysis on samples at a lower level of malignancy than PBZ and at a higher one: normal brain tissues, low-grade glioma (LGG), and GBM samples, respectively. Data from the The Cancer Genome Atlas (TCGA) and Genotype Tissue Expression (GTEx) database, consulted by OncoDB, showed that *CDK4* and *EXT2* mRNA expression increased with malignancy augmentation, rising from normal brain tissue, passing through LGG, ending in GBM (Figure 4). Indeed, the median expression values of *CDK4* and *EXT2* were 26.0 and 13.9 in brain tissue and they heightened up to 127.1 and 35.0, respectively, in GBM (Table 5). In addition, the expression levels of both genes was compared dividing the GBM cohort into the Verhaak’s molecular subtypes [19] (Appendix A). The only statistical difference revealed was between the mesenchymal and proneural subtypes in the *CDK4* expression (Appendix Aa).

Next, the prognostic value of *CDK4* and *EXT2* for LGG and GBM was further assessed in the TCGA and GTEx database cohorts through Gene Expression Profiling Interactive Analysis 2 (GEPIA2). No correlation seems to emerge with respect to overall survival, neither in GBM nor in LGG, if comparing the low-expression group and high-expression group of *CDK4* (Figure 5a). Nevertheless, dividing the GBM cohort into the Verhaak’s molecular subtypes [19], a trend for worse OS was observed in the proneural type with a higher expression of the gene (Figure 5a). Similarly, no differences were observed in the OS of GBM patients for *EXT2*, while in the LGG cohort the prognostic role of *EXT2*’s high expression took on different meanings during the observational period (Figure 5b). For *EXT2*, however, it is the Verhaak’s mesenchymal subtype, with increased gene expression, that shows decreased OS (Figure 5b). The co-overexpression of *CDK4* and *EXT2* affected the OS of LGG patients in a time-dependent way, similarly to *EXT2* alone, but not GBM ones; however, classical GBM patients had a significant worse overall survival in the case of co-overexpression of the two genes (Figure 5c).

Moreover, to suggest their potential combined function in inducing tumour transformation of the PBZ of GBM, the co-expression networks of both genes were analysed using the LinkedOmics database, focusing on TCGA-GBM data expression. The expression of 4741 genes and 2482 genes was positively and negatively related to *CDK4* in a significant manner; similarly, the expression of 7636 genes and 2574 genes was related to *EXT2* (Figure 6a). Heat maps were used to display the top 50 genes positively related to *CDK4* and *EXT2* (Figure 6b). Owing to the “comparing” function available on LinkedOmics, it was possible to compare the two analyses and finally to identify the 3965 overlapped genes positively related with both *CDK4* and *EXT2*.

Then, the gene set enrichment analysis (GSEA) was used to determine the main The Kyoto Encyclopedia of Genes and Genomes (KEGG) pathways enriched among the overlapped genes, and the results showed that they were mostly enriched in N-glycan biosynthesis, DNA replication, and proteasome (Figure 7).

Finally, to explore the potential mechanism by which *CDK4* and *EXT2* participated in tumour transformation of the peritumoural brain zone, the GeneMANIA online tool was used to construct a protein–protein interaction (PPI) network for both genes and the result is shown in Figure 8. As vividly shown in the figure, *CDK4* seemed to have multiple interactions with a larger number of genes than *EXT2*. Interestingly, both of them showed a direct and consistent physical interaction with *CDKN2A*, a tumour suppressor whose deletion is widely associated with GBM tumourigenesis [20].

## 3. Discussion

Tumour invasiveness, that makes surgery very challenging, and resistance to chemo-radiotherapy make glioblastoma one of the incurable and deadliest cancers. Emerging data showed that the peritumoural brain zone, the apparently healthy area surrounding the tumour core, is a key player in relapse occurrence, but still little is known about this not always accessible area. In fact, although the central role of PBZ in maintaining glioblastoma has been hypothesized, the collection of brain tissue poses ethical problems and surveys on large samples are not often possible [21,22,23,24]. 

In the past years, copy number alterations studies helped various groups to identify new tumour suppressor genes and oncogenes related to several kinds of cancers [25,26,27,28,29,30]. Although many studies have already widely investigated the genomic asset of the tumour core of GBM [11,26,31], a few of them focused on the genomic profile of the PBZ and on the steps leading to its malignant transformation [3,32]. 

In this work, the genomic comparison of the TC–PBZ pairs showed the presence of GBM CNAs in peritumoural margins, proving the infiltration of tumour cells in the PBZ parenchyma. Nevertheless, in the peritumoural brain zone we also found imbalances exclusive to this area, which may be involved in the early stages of the PBZ tumour transformation. With this in mind, in order to identify possible oncogenes, we have arbitrarily identified two genes involved in independent copy number gains, *CDK4* and *EXT2*, which could best play this role.

*CDK4* is widely known to be a cancer promoter gene and it is overactive in the majority of human cancers [33,34,35]. It regulates cell cycle entry forming, together with *CDK6* and one of the three D-type cyclins (D1, D2, and D3), the G1-S transition promoter complex that phosphorylates critical substrates, such as *RB1*, and promotes excessive cell proliferation in cancer [36]. In clinics, there are already drugs that specifically inhibit *CDK4*, such as palbociclib and abemaciclib, for the treatment of breast cancer [37,38]. In GBM, the Cyclin D1 (*CCND1*)–*CDK4/6*–*CDKN2A (*p16^INK4A^)–Rb axis is already known to be altered frequently [39,40,41,42]. Moreover, the potential role of *CDK4* in GBM is reinforced by the fact that both drugs are involved in clinical trials for glioblastoma treatments (DrugBank) [43]. *EXT2* is involved in the biosynthesis of the heparan sulphates (HSs), glycosaminoglycans distributed on the cell surfaces and in the extracellular matrices of most tissues [44]. HSs signalling was demonstrated to actively induce tumour proliferation, angiogenesis, metastasis, and differentiation, even in gliomas [45,46,47]. Nevertheless, *EXT2*’s role in cancer is still debated. In squamous cell lung carcinoma, Wu D. and colleagues observed its over-expression and classified it as a possible oncogene [17], while Ushakov and co-workers hypothesized its role as a tumour suppressor in human glioma [48]. 

Nowadays it is widely accepted that somatic CNAs influence cancer disease by impairing the expression level of cancer-related genes [49,50]. To verify if *CDK4* and *EXT2* were transcriptionally misbalanced in PBZ as being involved in CN gains, we performed gene expression analysis in 10 PBZ biopsies. Interestingly, even if the CN gains were not observed in all the samples, the majority of the biopsies evidenced over-expression of both genes and three biopsies had *EXT2* alone up-regulated. This highlights the need to use a multidisciplinary approach in studying heterogeneous disease such as GBM. In fact, non-highly sensitive techniques, such as array-based comparative genomic hybridization (array-CGH), cannot detect signatures of poorly represented cellular subpopulations, which are instead visible with quantitative PCR.

Assuming the possible role of *CDK4* in promoting the malignant transformation of normal brain tissue, we hypothesized that its over-expression in PBZ could affect patients’ survival. The survival analysis showed that patients with an increased expression of *CDK4* in the peritumoural margins had a statistically reduced overall survival, confirming our hypothesis. *EXT2* over-expression in almost the whole cohort of patients did not permit survival analysis, but its higher expression levels in the PBZ compared with the normal brain tissue could still correlate with the assumed pro-oncogenic role. 

The TCGA and GEtx databases through the GEPIA2 platform supported this hypothesis, showing increasing expression levels of both genes starting from normal brain tissues, passing through the LGG samples, and ending at the GBM ones. Even if *CDK4* and *EXT2* alone, or combined over-expression, did not affect the TCGA-GBM cohort, their unfavourable prognostic role emerged by stratifying the cohort in GBM molecular subtypes [19]. This data proved that the molecular characterization is necessary in order to classify GBM in subtypes by their molecular signatures otherwise hidden by the high heterogeneity of the disease [51,52]. Furthermore, these results, pointing out the cancer-promoting role of *CDK4* and *EXT2* specifically in GBM, allowed the contemplation of a similar function also in the PBZ. 

Taken together, the *CDK4*’s putative role as prognostic marker and the higher expression of *CDK4* and *EXT2* in PBZ samples we observed, together with the increasing expression of both genes in GBM compared with normal brain tissues, observed by consulting the databases, may suggest a driven role of both genes in the preliminary steps of PBZ malignant transformation.

Finally, the GSEA and the PPI analysis revealed the possibility that *CDK4* and *EXT2* could play the putative role together. Interestingly, *CDKN2A* seemed to interact directly with both *CDK4* and *EXT2*. *CDKN2A* loss is part of the canonical alterations found in the GBM landscape by Brennan and colleagues and subsequently confirmed by other works [11,53,54,55]. Its prognostic significance in making patients’ prognosis worse was proved both in LGGs and in GBMs [56]. In our samples, only four TC biopsies (from patients 9, 10, 17, and 38) and one PBZ biopsy (from patient 10) bear a deletion in *CDKN2A* locus (9p21.3), but alteration in *CDKN2A* expression cannot be excluded. 

Ultimately, in this work we assumed that *CDK4* and *EXT2* expression levels in the peritumoural brain zone may be halfway between those in the normal brain tissue and those in the GBM. In addition, their prognostic role in the TCGA cohort may suggest their involvement in the early malignant transformation of the normal parenchyma surrounding the tumour cavity and thus in the relapse occurrence.

These data lay the foundation for future investigations on the *CDK4* and *EXT2*’s role in the malignant transformation of the PBZ.

## 4. Materials and Methods

### 4.1. Study Population

A total of 28 patients that underwent surgical resection for GBM at San Gerardo Hospital (Monza, Italy) were enrolled in the study. Histological diagnosis was performed by a dedicated pathologist according to the 2021 WHO classification of central nervous system (CNS) tumours [57]. The study was approved by the ethics committee “Comitato Etico Monza e Brianza” (study number: 0031436—GLIODRUG-V, approved on 3 January 2020). The enrollment started in January 2020 and ended in December 2021. Demographic, clinical and immuno-phenotype data of patients and derived samples are reported in Table 1. Follow-up ended on 15 July 2022 (Table 1). Data were collected prospectively.

Tumour core biopsies were obtained through perilesional dissection during the surgical resection [58]. In order to collect non-neoplastic PBZ biopsies, the surgical cavity was checked with an intraoperative ultrasound (BK Medical, Herlev, Denmark) and samples were taken from the tumour-free surgical bed. More details are provided in Appendix A. PBZ samples were collected only from tissues considered far from functionally eloquent areas under neuronavigation (BrainLab, Munich, Germany) and ultrasound guidance (BK Medical, Herlev, Denmark). Biopsy samples were stored at −80 °C. 

The extent of surgical resection was measured according to Sanai et al., 2011, based on pre-operative and postoperative volumetric T1 weighted with gadolinium brain MRIs [59]. Gross total resection was considered when at least 99% of the tumour volume was removed, while subtotal resection was considered when tumour was removed from 98% to 80%. 

All patients were treated with concomitant chemo-radiation therapy according to Stupp’s protocol [60,61,62] and followed-up with a brain MRI every 3 months. Tumour recurrence was diagnosed using RANO criteria [62]. 

### 4.2. Immuno-Molecular Analysis

Immuno-molecular analysis was performed according to routine diagnostic procedures. Using the automated instrument Dako Omnis (Agilent Technologies, Santa Clara, CA, USA), immuno-histochemical staining was performed on 4% formalin-fixed paraffin-embedded sections of 1 µm thickness according to the manufacturer’s protocols. All antibodies were purchased from Dako Omnis (Agilent Technologies, Santa Clara, CA, USA). For p53, only nuclear staining was considered positive. MGMT promoter methylation was performed as previously reported [63]. Briefly, about 50–100 ng of genomic DNA obtained using automatic extraction (Maxwell, Promega, Madison, WI, USA) were subjected to bisulphite treatment using EZ DNA Methylation-Gold TM kit (Zymo Research, Irvine, CA, USA). The methylation status of six consecutive cytosines of MGMT promoter (chr10:131,265,507−131,265,556) was assessed by PCR-pyrosequencing of bisulphite-treated DNA by using MGMT Plus kit according to the recommended protocol (Diatech Pharmacogenetics, Jesi, Italy). A cut-off of 10% was set to score the presence of promoter methylation.

### 4.3. Genomic Analysis by Array-Based Genomic Comparative Hybridization

DNA were extracted from TC and PBZ biopsies and from patients’ blood (used as reference) using the automatic extractor iPrep TM (Thermo Fisher Scientific, Waltham, MA, USA) and the related kits supplied: iPrep tissue, for DNA extraction from biopsies; and iPrep whole blood, for DNA extraction from peripheral patients’ blood. Then, DNA were purified using Genomic DNA Clean & Concentrator kit (Zymo Research, Irvine, CA, USA). DNA quantity and quality were determined with NanoDrop ND-1000 Spectrophotometer (Thermo Fisher Scientific, Waltham, MA, USA). In cases in which the extracted DNA were not sufficient to perform further analysis, it was amplified using the GenomePlex Whole Genome Amplification (WGA) Kit (Sigma-Aldrich, St. Louis, MI, USA) according to the manufacturer’s instructions. Amplified DNA were tested for purity and concentration as above.

Array-based Comparative Genomic Hybridization (array-CGH) analysis was performed using 60 mer oligonucleotide probe technology (SurePrint G3 Human CGH 8 × 60 K, Agilent Technologies, Santa Clara, CA, USA) according to the manufacturer’s instructions. Agilent Feature Extraction was exploited to generate raw data, which were further analyzed using Cytogenomics 5.1 with the ADAM-2 algorithm (Agilent Technologies, Santa Clara, CA, USA). A minimum of three consecutive probes/regions were considered as a filter. The threshold for genomic deletion is x = −1; the threshold for genomic gain is x = +0.58. Amplifications and homozygous deletions were considered with threshold >+2 and <−1, respectively.

### 4.4. Strategies Adopted to Select Interesting Genes of the PBZ Involved in Copy Number Alterations

#### 4.4.1. Selection of CNAs

Two strategies were applied in order to select CNAs specific to the PBZ. Strategy I: firstly comparing the CNAs of each matched TC-PBZ pair, then selecting CNAs present only in the PBZ and shared between at least two different patients. Strategy II: selecting CNAs present in the PBZ of at least 4 different patients, and not necessarily absent in the TCs (Appendix A). 

#### 4.4.2. Selection of Genes

Information about genes included in CN gains that emerged from the first phase was collected by consulting both literature [16,17,18] and specific databases (The Kyoto Encyclopedia of Genes and Genomes database—KEGG database, Human Protein Atlas—HPA, and DrugBank, see 4.6 Databases section for further information). Those genes related with cancer pathways and with potential oncogenetic properties, already known to be associated with tumours, and druggable, were finally selected.

### 4.5. TaqMan Gene Expression Assay

RNA was extracted using the RNeasy Mini-Kit (Qiagen, Hilden, Germany) according to the manufacturer’s protocol from tumour core and peritumour biopsies and from normal human astrocytes (NHA) cells (Lonza, Morrisville, NC, USA). NHA cells were kindly provided by Professor Francesca Re (Medicine and Surgery Department, University of Milano-Bicocca, Monza, Italy). RNA quantity and quality were determined with a Nanodrop ND-2000 spectrophotometer (Thermo Fisher Scientific, Waltham, MA, USA).

RNA samples were converted into first-strand cDNA using the High Capacity cDNA Reverse Transcription Kit (Thermo Fisher Scientific, Waltham, MA, USA) following the manufacturer’s instructions. 

TaqMan gene expression assays (Applied Biosystems, Waltham, MA, USA) were performed in order to evaluate the expression levels of *CDK4* and *EXT2* in the peritumoural brain zone biopsies. *GAPDH* was used as a housekeeping gene, while NHA cells were used as normal control. All the probes used were purchased from Applied Biosystems (Applied Biosystems, Waltham, MA, USA). Quantitative PCR was carried out using the ABI StepOne Plus (Applied Biosystems, Waltham, MA, USA) according to the manufacturer’s instructions. The cycle conditions were as follows: 10 min 95 °C; 40 cycles: 15 s 95 °C; and 1 min 60 °C. The comparative threshold cycle (Ct) method, the 2^−ΔΔCt^ method, was used to calculate the fold amplification of each gene in PBZs compared with the normal control (NHA), set at 1. It was not possible to replicate the analysis since the available biological samples were limited.

### 4.6. Databases

#### 4.6.1. Selection of *CDK4* and *EXT2*

The Kyoto Encyclopedia of Genes and Genomes (KEGG) database (https://www.genome.jp/kegg/pathway.html, Version 103.1, Kyoto University, Kyoto, Japan, last access date: 10 September 2022) is a bioinformatics resource for mining significantly altered metabolic pathways enriched in the gene list. It was consulted to evaluate in which pathways the selected genes were involved. 

The Human Protein Atlas (https://www.proteinatlas.org, Version 21.1, last access date: 30 October 2022) was consulted to evaluate the tissue expression of the selected genes and their role as prognostic markers in cancers.

The DrugBank database (https://go.drugbank.com, Version 5.0, University of Alberta and The Metabolomics Innovation Centre located, Alberta, Canada, last access date: 10 September 2022) [43] is a richly annotated resource that combines detailed drug data with comprehensive drug target and action information. It was consulted to evaluate the presence of compounds targeting the selected genes.

#### 4.6.2. Analysis of *CDK4* and *EXT2* Expression in Low-Grade Glioma and Glioblastoma

The RNA sequencing expression data from the The Cancer Genome Atlas (TCGA) and the Genotype Tissue Expression (GTEx) projects were used to determine *CDK4* and *EXT2* expression in tumours (low-grade glioma—LGG and GBM) and normal brain tissues, consulting the OncoDB database (https://oncodb.org/index.html, Version 1.0, last access date: 13 January 2023) [64].

Gene Expression Profiling Interactive Analysis 2 (GEPIA2) (http://gepia.cancer-pku.cn, Zhang Lab, Peking University, Beijing, China, last access date: 30 October 2022) [65] is an online platform that was used in order to evaluate the *CDK4* and *EXT2* expression among the molecular subtypes of GBM from the RNA sequencing expression data of the TCGA cohort.

RNA sequencing data were analysed using the parameter log10 (TPM+1) (transcripts per kilobase of exon model per million mapped reads).

#### 4.6.3. Survival Analysis and Relationship with GBM Molecular Subtypes

The GEPIA2 (http://gepia.cancer-pku.cn, Zhang Lab, Peking University, Beijing, China, last access date: 30 October 2022) [65] database is an online platform for dissecting the RNA sequencing expression data from the TCGA and the GTEx projects, using a standard processing approach. The “Survival” module of GEPIA2 was used to assess the correlation between *CDK4* and *EXT2* expression and prognosis of LGG and GBM patients. GEPIA2 also provided interactive functions such as profiling according to molecular subtypes of GBM. 

#### 4.6.4. *CDK4* and *EXT2* Co-Expression Networks

The LinkedOmics database (http://www.linkedomics.org, Zhang Lab, Baylor College of Medicine, Huston, TX, USA, last access date 13 November 2022) is a visual platform used to explore the gene expression profile [66]. LinkedOmics was used to determine the *CDK4* and *EXT2* co-expressed genes through Pearson’s correlation coefficient. The results were shown via heat maps. Then, the KEGG pathways of the positively related genes of both *CDK4* and *EXT2* were explored by using gene set enrichment analysis (GSEA).

#### 4.6.5. Protein–Protein Interaction Network Construction

The GeneMANIA prediction server: biological network integration for gene prioritization and predicting gene function (http://www.genemania.org, Toronto University, Toronto, Canada, last access date: 11 November 2022) is an interactive website for constructing protein–protein interaction (PPI) networks, which generates hypotheses about gene function prediction and detects genes with similar functions [67]. This network integration algorithm features the following bioinformatics methods: physical interaction, co-expression, co-localization, gene enrichment analysis, genetic interaction, and website prediction. In this study, GeneMANIA was applied for PPI analysis of *CDK4* and *EXT2*.

### 4.7. Statistics

Disease-free survival (DFS) and overall survival (OS) were estimated by the Kaplan–Meier method and comparison of curves was performed using a log-rank test; hazard ratio, 95% confidence interval, and log-rank *p* values were displayed. Statistical analyses were carried out performing Fisher’s exact chi-square test. Pearson correlation was used to compare the genomic profiles in matched pairs of TCs and related PBZ samples. The statistical significance for each pair of correlations was calculated by consulting the Table of Critical Values for Pearson’s R. The level of significance for a two-tailed test was set to α = 0.05. Gene expression data from the TCGA and GTEx databases were analyzed using one way ANOVA and Student’s *t*-test. *p* value < 0.05 was considered statistically significant. The false discovery rate (FDR) was used as the cut-off in the enrichment analysis.

## 5. Conclusions

The main limitation of this study is the small size of the sample cohort. This fact leads to clinical results in contrast to literature. For example, several studies have proved the positive effect of the gross total resection on disease free survival in gliomas [68,69]. Even if about 71% of our patients received a GTR, a result in line with the literature data [70,71], no significant differences in terms of disease-free survival between patients almost completely resected and those who received a sub-total resection emerged. On the other hand, in our work, among the risk factors for poor prognosis, age onset influenced patients’ outcome, which is in line with the current literature data [72]. Meanwhile, the methylation status of the MGMT promoter did not result to be associated with overall survival, even if it normally correlates with patients’ outcome and with a positive response to temozolomide treatment [73]. The small sample size of our series of patients and the fact that nearly half of the patients were still alive at the end of the follow-up probably caused these findings. Besides these considerations, it is necessary to highlight that MGMT is not the only factor that can affect patients’ response to temozolomide; other mechanisms such as repairing DNA damage induced by chemotherapy [74,75], hindering the drug delivery [76], or acidifying the tumour microenvironments [12,13,77] are implemented by the tumour. For this reason, even if in our study both *CDK4* and *EXT2* were found over-expressed in almost the whole cohort, it was mandatory to integrate the results with databases although they are focused on normal or tumoural tissues and not on PBZ, an area that is halfway to the malignancy.

In conclusion, this preliminary study pointed out for the first time, to the best of our knowledge, the potential role of *CDK4* and *EXT2* in the peritumoural brain zone of glioblastoma as driver genes forwarding the malignant transformation of this area. Although our data require further validation, they could be of interest for those groups involved in glioma or cancer research. Our future goal is to consolidate our findings in further studies with a larger sample size and a longer clinical and radiological follow-up in order to strengthen our preliminary results. If confirmed, the outcome may lead to finding druggable targets that may increase disease control and patients’ survival.

## Figures and Tables

**Figure 1 ijms-24-02835-f001:**
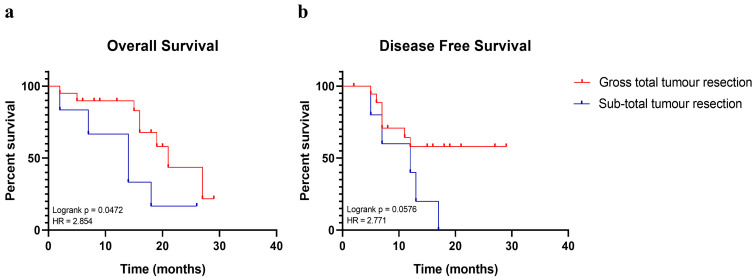
Kaplan–Meier curves performed on patients enrolled in the study. Survival analysis displays the estimated survival of patients who either received a gross total tumour resection during surgery (in red) or who received a sub-total one (in blue). Timing is expressed in months. (**a**) Shows the effects of the gross total tumour resection on the overall survival. (**b**) Shows the effects of the gross total tumour resection on the disease-free survival. Log-rank *p* values and hazard ratio (HR) are reported.

**Figure 2 ijms-24-02835-f002:**
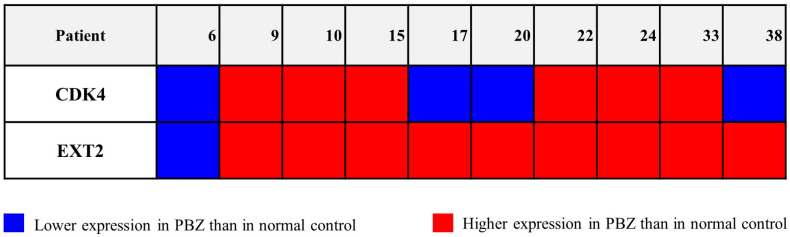
*CDK4* and *EXT2* expression levels in the peritumoural biopsy (PBZ) of each patient. Blue, indicates a lower expression of the target gene in the PBZ compared with the control; red, indicates a higher expression of the target gene in the PBZ compared with the control. Normal human astrocytes (NHA) cells are used as the control.

**Figure 3 ijms-24-02835-f003:**
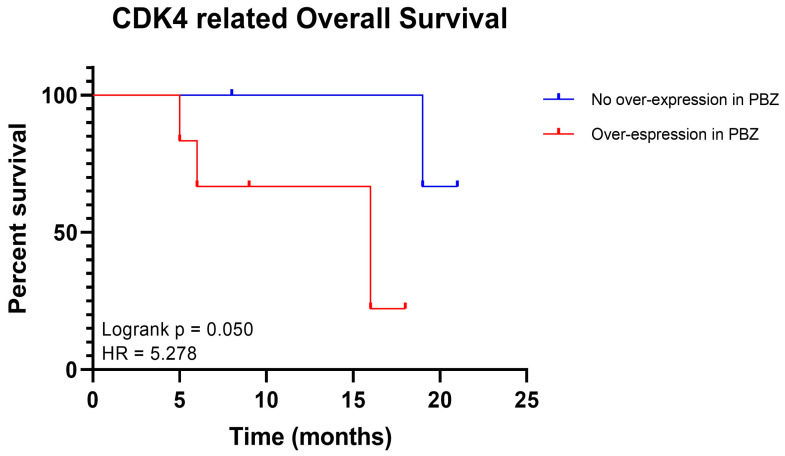
Kaplan–Meier curves performed on patients enrolled in the study. Survival analysis displays the estimated survival for patients without the over-expression of *CDK4* in their peritumoural brain zone (PBZ) (in blue) or patients with the over-expression of *CDK4* in their PBZ (in red). Timing is expressed in months. Log-rank *p* values and hazard ratio (HR) are reported.

**Figure 4 ijms-24-02835-f004:**
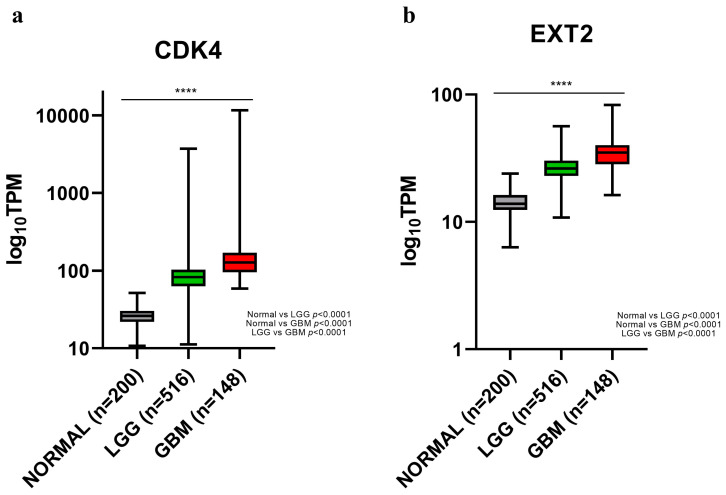
Differential *CDK4* (**a**) and *EXT2* (**b**) mRNA expression among normal brain tissue, low-grade glioma (LGG), and glioblastoma (GBM) in the OncoDB database. Grey boxes refer to normal brain tissue, green boxes refer to LGG, and red boxes refer to GBM. TPM: transcripts per kilobase of exon model per million mapped reads. **** *p* < 0.0001 (one way ANOVA test); *p* value of each comparison are reported.

**Figure 5 ijms-24-02835-f005:**
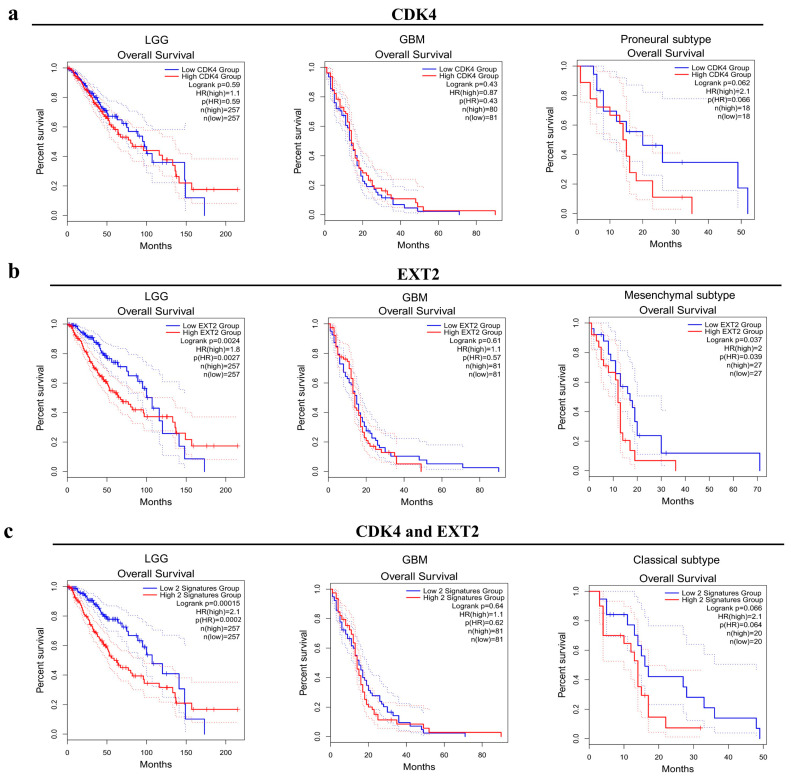
Kaplan–Meier curves of the overall survival of low-grade glioma (LGG) and glioblastoma (GBM) TCGA cohorts by *CDK4* alone (**a**), *EXT2* alone (**b**), *CDK4* and *EXT2* (**c**) mRNA expression, via GEPIA2. In blue, survival curves of patients with a low expression of the target gene(s); in red, survival curves of patients with a high expression of the target gene(s). Timing is expressed in months. Log-rank *p* values, hazard ratio (HR), HR *p* values, and the number of participants (n) of each group are reported.

**Figure 6 ijms-24-02835-f006:**
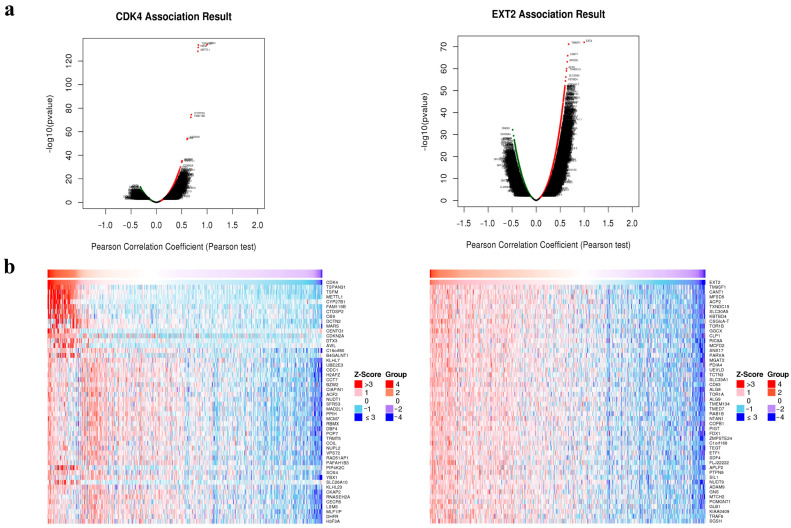
Co-expression networks of *CDK4* and *EXT2*. (**a**) Volcano plots show highly correlated genes of *CDK4* (left) and *EXT2* (right) tested by Pearson test in the glioblastoma (GBM) cohort by the LinkedOmics database. Dark red dots indicate genes positively correlated; dark green dots indicate genes negatively correlated. (**b**) Top 50 positive co-expressed genes of *CDK4* (left) and *EXT2* (right) in heat maps in the GBM cohort by the LinkedOmics database.

**Figure 7 ijms-24-02835-f007:**
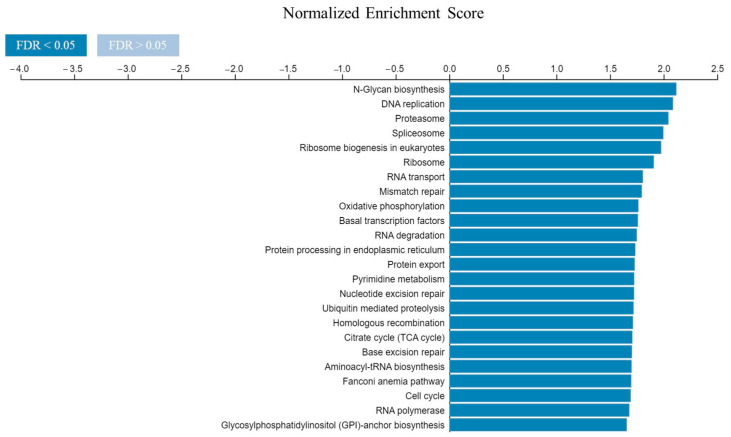
Gene set enrichment analysis (GSEA) of the KEGG pathways mostly enriched among genes positively related with both *CDK4* and *EXT2*, performed by LinkedOmics database. The colour of the bars indicates the values of false discovery rate (FDR).

**Figure 8 ijms-24-02835-f008:**
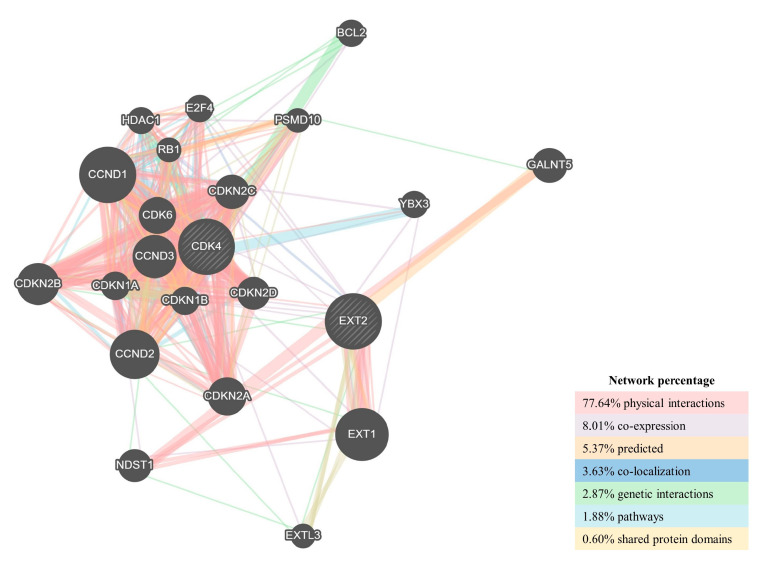
Protein–protein interaction (PPI) network for *CDK4* and *EXT2* was constructed in GeneMANIA. Different colours of the network edge indicate the bioinformatics methods applied: physical interaction, co-expression, predicted, co-localization, pathway, genetic interaction, and shared protein domains.

**Table 1 ijms-24-02835-t001:** Clinic–pathological characteristics, immuno-molecular phenotype, details on the type of surgery performed (gross total tumour resection—GTR), and disease monitoring and surveillance of patients enrolled in the study are reported. M: male; F: female; GBM: glioblastoma IDH-wildtype; ^+^: >30% of positive cells; ^−^: <30% of positive cells; met: methylated; wt: wildtype; DFS: disease-free survival; OS: overall survival; -: no information; PE: pulmonary embolism.

Patient	Sex	Age	Tumour Location	Immuno-phenotype	GTR	DFS (Months)	OS (Months)	PBZ Biopsy
1	M	71	Right parietal lobe	ATRX^+^, p53^−^, MGMT met	No	12	14	No
2	M	54	Right frontal lobe	ATRX^−^, p53^+^, MGMT met	Yes	No progression	27	No
3	F	61	Left temporal lobe	ATRX^+^, p53^−^, MGMT met	Yes	No progression	Alive (29)	No
5	M	57	Left temporal lobe	ATRX^+^, p53^−^, MGMT wt	Yes	No progression	21	No
6	M	44	Right temporal lobe	ATRX^+^, p53^−^, MGMT wt	Yes	11	19	Yes
7	M	54	Right parietal lobe	ATRX^+^, p53^−^, MGMT wt	Yes	6	15	No
8	M	57	Right temporal lobe	ATRX^+^, p53^−^, MGMT met	No	17	Alive (26)	No
9	M	71	Left occipital lobe	ATRX^+^, p53^−^, MGMT wt	Yes	12	16	Yes
10	M	74	Right temporo- occipital lobe	ATRX^+^, p53^−^, MGMT met	Yes	No progression	5	Yes
11	M	76	Right temporal lobe	ATRX^+^, p53^−^, MGMT met	No	13	18	Yes
12	F	70	Multifocal	ATRX^+^, p53^−^, MGMT wt	Yes	<1	2	No
13	M	58	Left temporal lobe	ATRX^+^, p53^−^, MGMT met	No	7	14	No
14	M	73	Left temporal lobe	ATRX^+^, p53^−^, MGMT met	-	No progression	<1 (died for PE)	No
15	M	78	Left frontal lobe	ATRX^+^, p53^−^, MGMT met	Yes	No progression	16	Yes
17	M	81	Right frontal lobe	ATRX^+^, p53^−^, MGMT wt	Yes	No progression	Alive (21)	Yes
18	F	49	Right occipital lobe	ATRX^+^, p53^−^, MGMT wt	Yes	7	Alive (20)	No
20	F	60	Right occipital lobe	ATRX^+^, p53^−^, MGMT wt	Yes	No progression	Alive (19)	Yes
21	M	39	Right frontal lobe	ATRX^−^, p53^+^, MGMT met	Yes	No progression	Alive (18)	No
22	M	53	Left temporal lobe	ATRX^+^, p53^−^, MGMT met	Yes	No progression	Alive (18)	Yes
23	M	71	Left frontal lobe	ATRX^+^, p53^−^, MGMT met	No	5	7	No
24	M	69	Left parieto- occipital lobe	ATRX^+^, p53^−^, MGMT wt	Yes	5	6	Yes
26	F	69	Left parietal lobe	ATRX^+^, p53^−^, MGMT met	Yes	No progression	Alive (15)	No
27	M	57	Right temporal lobe	ATRX^+^, p53^−^, MGMT wt	No	<1	2	Yes
28	M	59	Right frontal lobe	ATRX^+^, p53^−^, MGMT wt	Yes	No progression	Alive (2)	Yes
30	F	73	Right frontal lobe	ATRX^+^, p53^−^, MGMT met	-	No progression	<1 (died for ventriculitis)	Yes
31	M	63	Left temporo- occipital lobe	ATRX^+^, p53^−^, MGMT wt	Yes	7	Alive (12)	Yes
33	M	54	Right temporo-occipital lobe	ATRX^−^, p53^−^, MGMT met	Yes	7	Alive (9)	Yes
38	M	49	Right parieto-occipital lobe	ATRX^+^, p53^−^, MGMT wt	Yes	No progression	Alive (8)	Yes

**Table 2 ijms-24-02835-t002:** Report of Pearson correlation values (R) for tumour core–peritumoural brain zone genomic profiles comparison. Patients’ pairs of biopsies were ranked into two groups: ones with a moderately strong correlation (0.40 ≤ R < 0.80); ones with a low correlation (R < 0.40). The statistical significance for each pair of correlations was calculated consulting the Table of Critical Values for Pearson’s R. N: total number of copy number alterations; DF: degree of freedom (N-2). Level of significance for a two-tailed test α = 0.05. * indicates statistically significant test. # shows data published in [11].

Patient	R	N	DF	Critical Value of R	Correlation
10 #	0.62	71	69	0.232 *	Moderately strong
17 #	0.42	24	22	0.404 *	Moderately strong
15 #	−0.87	24	22	0.404 *	Low
22 #	−0.52	23	21	0.413 *	Low
24 #	0.37	36	34	0.325 *	Low
28	−0.74094	13	11	0.553 *	Low
30	0.226147	23	21	0.413	Low
31	0.061161	37	35	0.325	Low
38	0.671874	19	17	0.456 *	Moderately strong

**Table 3 ijms-24-02835-t003:** Selection of copy number alterations (CNAs) by the two applied strategies (I and II). Associated patients in whom they are found and the list of genes affected by each imbalance are reported.

	CNAs Locus	Patients	List of Genes Affected
Strategy I	11p11.2 Gain	9 22 28	*EXT2*
	16p13.3 Gain	15 22	*SSTR5-AS1, SSTR5, C1QTNF8, CACNA1H, TPSG1, TPSB2*,*TPSAB1*,*TPSD1*
Strategy II	7q21.12 Gain	11 20 22 27 38	*GRM3*, *KIAA1324L*
	10q21.3 Loss	11 20 22 27	*DNA2*, *SLC25A16*, *TET1*,*CCAR1*, *SNORD98*, *STOX1*, *DDX50*,*DDX21*, *KIF1BP*, *SRGN*, *VPS26A*, *SUPV3L1*, *LOC101928994*, *HKDC1*
	12q14.1 Gain	10 22 24 27	*AGAP2*, *AGAP2-AS1*, *TSPAN31*, *CDK4*, *MIR6759*, *MARCHF9*, *CYP27B1*, *METTL1*, *EEF1AKMT3*, *TSFM*, *AVIL*, *CTDSP2*
	9p21.3 Loss	10 11 17 20 22	*CDKN2A-DT*, *CDKN2A*, *CDKN2B-AS1*, *CDKN2B*

**Table 4 ijms-24-02835-t004:** Selection of genes involved in copy number gains after consulting the literature and databases (The Kyoto Encyclopedia of Genes and Genomes database—KEGG database, Human Protein Atlas - HPA, and DrugBank).

Gene	Cancer-Related Pathway	Associated Tumours	Prognostic Marker	Targeting Compound
*CDK4*	Cell cycle, p53 signalling pathway, PI3K-Akt signalling pathway, cellular senescence (KEGG database)	Metastatic breast cancer, bladder, head and neck squamous cell carcinoma, oesophageal [16]	Unfavourable in renal and liver cancers (HPA)	Alvocidib, purvalanol, palbociclib, ribociclib, abemaciclib, fostamatinib (DrugBank)
*EXT2*	Glycosaminoglycan biosynthesis–heparan sulphate/heparin, metabolic pathways (KEGG database)	Squamous cell lung carcinoma [17], head and neck squamous cell carcinoma [18]	Unfavourable in renal and head and neck cancers (HPA)	Uridine-diphosphate-N-acetylgalactosamine, uridine-5′-diphosphate (DrugBank)

**Table 5 ijms-24-02835-t005:** Median expression of *CDK4* and *EXT2* mRNAs in normal brain tissue, low-grade glioma (LGG), and glioblastoma (GBM) in the OncoDB database, expressed in transcripts per kilobase of exon model per million mapped reads.

Type of Sample	*CDK4*		*EXT2*	
	Median Expression	Log2 (Fold Change)	Median Expression	Log2 (Fold Change)
Normal brain tissue	26.0		13.9	
LGG	82.5	1.67	26.3	0.92
GBM	127.1	2.29	35.0	1.32

## Data Availability

The data used to support the findings of this study are available from the corresponding author upon request.

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
