# Peer review of "Insights into the Peritumoural Brain Zone of Glioblastoma: CDK4 and EXT2 May Be Potential Drivers of Malignancy"

_ijms, 2023, doi:10.3390/ijms24032835_

Round 1
Reviewer 1 Report
The manuscript “Insights the peritumoral brain zone of glioblastoma: CDK4 2 and EXT2 may be potential drivers of malignancy” by Giambra M. and Colleagues is the follow-up of a previous work published in 2021 in “Biology”. The aim of this work is to identify news therapeutics targets by pair comparison of tumor core (TC) versus peritumoral border zone (PBZ) from glioblastoma patients. By paired genomic analysis of TC and PBZ from 15 patients and using a bioinformatic approach (several databases are investigated), the authors focused their attention on two genes presenting copy number gain alterations which may play a role in tumor progression. The question is of interest for the field and access to clinical data or patient material are an asset for the study. However, the results are not clearly presented and do not support the conclusions. Indeed, authors do not present results on the functional implication of Cyclin-Dependent Kinase-4 (CDK4) and Exostosin glycosyltransferase-2 (EXT2) in glioblastoma progression even though this is the claim of the title. The work is preliminary and cannot be published in its present form.
Major comments
1. More details are needed on the way the PBZ has been defined for each patient sample. Is it based on distance from TC or on histological features? What are criteria used to compare several PBZ from several patients? PBZ histology or imaging of PBZ before and after resection must be shown.
2. Informations about the patient cohort are not clearly provided and must be better described. The study was made on 15 out of 28 patients for paired TC and PBZ analysis (table 1). Among the 15 patients with PBZ biopsy only 7 had GTR (Gross Tumor resection, table 1+2). It is not clear what is the link between GRT, overall survival and PBZ genetic alterations. The number of samples is by far too low to have solid statistical results.
3. Figure 2. Authors show the level of mRNA expression of CDK4 and EXT2 in PBZ compared to normal control. Data from protein analysis would be relevant, in particular from immunohistological analysis or FACS analysis. This last technique would allow to characterize the cell populations in PBZ expressing these molecules. Normal controls are not defined.
4. Figure 4. Data should be statistically analysed to determine the significant difference between mRNA expression from Normal, LGG and GBM TCGA data. Authors show a significantly difference between Normal vs LGG or normal vs GBM. Is the difference significant between LGG and GBM? A two way Anova analysis must be provided.
5. The functional role of CDK4 and EXT2 must be further analysed by genetic or pharmacologic approaches. References to previously published work in glioblastoma is not provided and must be cited.
6. Figure 5. The authors provide OS curves according to expression of CDK4 and EXT2 on GBM subtypes. However, the expression in the different subtypes is not analysed.
7. Figure 6,7 and 8 are rather correlative and do not add strengths to the results.
8. Figure 9: the model proposed is not supported by the results.
Minor comments
Material an method section must be more detailed.
The results are not sufficiently described: what is the purpose of the analysis, what are the conclusion for each result section?
Table 3. All patients analyzed should be included in the table.

Author Response
Response to Reviewer 1 Comments
- More details are needed on the way the PBZ has been defined for each patient sample. Is it based on distance from TC or on histological features? What are criteria used to compare several PBZ from several patients? PBZ histology or imaging of PBZ before and after resection must be shown.
R: We thank the reviewer for the opinions. As described in the Material and Methods section, the collection of PBZ samples was performed at the end of tumor resection by expert and dedicated neuro-oncological neurosurgeons. After TC resection ended, the surgical cavity was checked with an intraoperative US in order to find brain parenchyma far from TC and not infiltrated by frankly tumoral brain tissue. We defined this area as PBZ thanks to the further histological analysis: PBZ was considered negative for tumor infiltration when there was a normal cellularity in comparison with normal brain parenchyma and when it was not possible to identify neoplastic cells similar to those found in the TC. The lack of standardized and well defined criteria and markers for PBZ definition, as well as the absence of an official WHO consensus on this issue, prompted us to compare several samples from several patients in order to better describe this specific area. With our studies we hope to help to fill the gap about the PBZ knowledge and we hope to contribute to find a common definition of this histological area, as we underlined in our recent Review (ref 3). We added in the supplementary materials an example of PBZ’s histology and MRI images before and after surgery and their description.
- Informations about the patient cohort are not clearly provided and must be better described. The study was made on 15 out of 28 patients for paired TC and PBZ analysis (table 1). Among the 15 patients with PBZ biopsy only 7 had GTR (Gross Tumor resection, table 1+2). It is not clear what is the link between GRT, overall survival and PBZ genetic alterations. The number of samples is by far too low to have solid statistical results.
R: We thank the reviewer for sharing their point of view. We modified the two tables joining them together, in order to avoid misunderstanding (now named table 1). We agree with the reviewer about the limited size of our cohort, as we claimed in the Conclusions (section 5). We believe that this preliminary study could help to fill the gap in PBZ knowledge and we are working in order to provide solid statistical results in the future enlarging the sample size.
- Figure 2. Authors show the level of mRNA expression of CDK4 and EXT2 in PBZ compared to normal control. Data from protein analysis would be relevant, in particular from immunohistological analysis or FACS analysis. This last technique would allow to characterize the cell populations in PBZ expressing these molecules. Normal controls are not defined.
R: We thank the reviewer for the comment. We are aware of the limitations of this study and that a protein analysis could be relevant; however, since these are preliminary data and on few samples, we are currently trying to extend the number of samples to confirm the data and move on to protein investigations that require a greater economic effort. As we pointed out in the discussion, recruiting these samples from the margin is not always feasible, so we believe that our results could help researchers discover novel tumorigenic mechanisms associated with PBZ. We specified that Normal Human Astrocytes (NHA) cells were used as control both in the caption of the figure 2 and in the main text (see Results section 2.2.).
- Figure 4. Data should be statistically analysed to determine the significant difference between mRNA expression from Normal, LGG and GBM TCGA data. Authors show a significantly difference between Normal vs LGG or normal vs GBM. Is the difference significant between LGG and GBM? A two way Anova analysis must be provided.
R: We thank the reviewer for their hint. In order to satisfy their will, we used OncoDB (https://oncodb.org/index.html) instead of GEPIA2, which allows us to download the TCGA raw data and perform statistical analysis. We integrated the obtained results in the main text (see Result section 2.3). We performed the one way ANOVA tests in order to compare the expression level of each gene among the three groups (Normal, LGG and GBM), and the Student’s t-tests to highlight the difference between Normal vs LGG, Normal vs GBM and LGG vs GBM. Furthermore, we added the OncoDB database in the dedicated section of Material and Methods.
- The functional role of CDK4 and EXT2 must be further analysed by genetic or pharmacologic approaches. References to previously published work in glioblastoma is not provided and must be cited.
R: We thank the reviewer for this comment. As we claimed in response 3, we are currently trying to extend the number of samples to confirm the data in order to consolidate the rationale prior to moving towards further analysis. We added the required references in the discussion. As concern CDK4 references 39-42, while for EXT2 45-48.
- Figure 5. The authors provide OS curves according to expression of CDK4 and EXT2 on GBM subtypes. However, the expression in the different subtypes is not analysed.
R: We thank the reviewer for the suggestion. The RNA sequencing expression data from the TCGA cohort was analysed using GEPIA2 and the CDK4 and EXT2 expression levels among GBM molecular subtypes are now provided as well (see Result section 2.3 and figure S1).
- Figure 6,7 and 8 are rather correlative and do not add strengths to the results.
R: We thank the reviewer for this comment. We decided to integrate the analysis about CDK4 and EXT2 expression, performed in our cohort of samples and in TCGA’s and GTEx’s ones, together with the results obtained from LinkedOmics and GeneMANIA to execute a comprehensive survey of the potential role of the genes in GBM. Since our data are not solid statistically due to the limited dimension of the sample, we wanted to take advantage of the abundant bioinformatics database available today to support the hypothetical model proposed here. Furthermore, in the last years, many studies have benefited from databases; some of them actually based their entire results on the bioinformatics analysis (Feng S. Front Oncol. 2022, Li X. Am J Transl Res. 2022, Zhou X. Front Oncol. 2022, Hoda A. J Biomol Struct Dyn. 2013, Qiu T. Front Genet. 2022).
- Figure 9: the model proposed is not supported by the results.
R: We thank the reviewer for sharing their opinion with us. Although it was not our will proposing here a definitive model, we agree with your point. For this reason, we deleted figure 9 and we are working on further analysis in order to validate our assumptions and to find stronger evidence to support our model.
Minor comments
- Material an method section must be more detailed.
R: We thank the reviewer for the suggestion. We reviewed the Material and Methods section and we provided more details where needed.
- The results are not sufficiently described: what is the purpose of the analysis, what are the conclusion for each result section?
R: We thank the reviewer for the comment. We added or better specified what the purposes and the conclusions were in each Result section.
- Table 3. All patients analyzed should be included in the table.
R: We included in the table (now named table 2) the data already published in our previous work.
Reviewer 2 Report
This manuscript compared the genomic profiles of matched tumor core and peritumoral brain zone (PBZ) biopsies, to pick CDK4 and EXT2 as potential oncogenes of PBZ, which are supported by further bioinformatic analysis of TCGA data and protein-protein interactions. My concerns for this paper are the following:
(1) The author picks these two genes arbitrarily, which makes studying their co-effect pointless.
(2) Studying CDK4 and EXT4 based on the TCGA database lacks novelty, which is already reported by other groups.
(3) The overall picture resolution is poor.
Author Response
Response to Reviewer 2 Comments
1. The author picks these two genes arbitrarily, which makes studying their co-effect pointless.
R: We thank the reviewer for their points and suggestions. It was necessary in this preliminary study to arbitrarily select strategies that would help us to manage and sort the huge amount of data obtained by the genomic analysis that was our starting point (ref. 11). Moreover, since little is known about the PBZ, we could not base our decisions on previous works, so we picked CDK4 and EXT2 because they were the candidates that better met the selection requirements chosen (see Materials and Methods section 4.4). Nevertheless, we already have started working and investigating other possible genes involved in PBZ and emerged from the genomic analysis (ref. 11).
2. Studying CDK4 and EXT4 based on the TCGA database lacks novelty, which is already reported by other groups.
R: We thank the reviewer for the comment. We agree about the fact that studying CDK4 and EXT2 based on the TCGA database lacks novelty. We decided to consult TCGA and GTEx databases to collect data about LGG and GBM and normal brain tissue, respectively, since databases focused on the peritumoral brain zone, the topic of this preliminary work, are still absent. We thought that in order to give solidity to the results obtained from our limited samples of PBZ biopsies, we should have reported what happens in the previous (normal brain tissue) and subsequent (tumours) grade of malignancy, as PBZ is halfway between them. Our future perspective is to enlarge the number of PBZ samples to create a valid casuistry on a biological area still not completely known.
3. The overall picture resolution is poor.
R: We thank the reviewer and we improved the picture resolution to 300 dpi.
Round 2
Reviewer 1 Report
I thank the authors having made the effort to add more clarity on the results section. Table 1, table 2, and figure 4 are now more comprehensive and Suppl. Figure S1 and S3 add some information. Moreover, I appreciated their effort to add more details in the material and method section. However, even if the question is of real interest for the field, data remain preliminary and the presentation of the results remain still confused and difficult to follow. The work to my concern remain still preliminary and cannot be published in its present form.
Minor comments: figure 2 and figure 5 appear now twice in the manuscript
Reviewer 2 Report
The authors have cleared my concerns.